# Guideline adherence in the management of head injury in Australian children: A population-based sample survey

Janet C. Long[1], Sarah Dalton[2,3], Gaston Arnolda[1], Hsuen P. Ting[1], Charlotte J. Molloy[1,4], Peter D. Hibbert[1,4,5], Louise K. Wiles[1,4,5], Simon Craig[6], Meagan Warwick[1], Kate Churruca[1], Louise A. Ellis[1], Jeffrey Braithwaite[1]*, on behalf of the CareTrack Kids investigative team¶

1 Australian Institute of Health Innovation, Macquarie University, Sydney, New South Wales, Australia,
2 Emergency Department, The Children's Hospital at Westmead, Sydney, New South Wales, Australia,
3 Agency for Clinical Innovation, Sydney, New South Wales, Australia, 4 Australian Centre for Precision Health, University of South Australia Cancer Research Institute, University of South Australia, Adelaide, South Australia, Australia, 5 South Australian Health and Medical Research Institute (SAHMRI), Adelaide, South Australia, Australia, 6 Department of Paediatrics, School of Clinical Sciences at Monash Health, Monash University, Melbourne, Victoria, Australia

¶ Membership of the author group can be found in the Acknowledgments.
* jeffrey.braithwaite@mq.edu.au

**Data Availability Statement:** Patient data in this study are not publicly available as they were collected from medical records examined by the research team without seeking individual consent.

## Abstract

### Background

Head injuries in children are a common and potentially devastating presentation. The Care-Track Kids (CTK) study assessed care of Australian children aged 0–15 years, in 2012 and 2013, to evaluate the proportion in line with guideline-based indicators for 17 common conditions. Overall adherence to guideline-based recommended practice occurred 59.8% of care encounters (95% CI: 57.5–62.0), and 78.3% (95% CI: 75.1–81.2) for head injury. This paper presents results for head injury, at indicator level.

### Methods

A modified version of the RAND-UCLA method of indicator development was used. Indicators, measurable components of a standard or guideline, were developed from international and national guidelines relating to head injury in children and were ratified by clinical experts using a Delphi process. Paediatric nurses extracted data from medical records from general practitioners (GPs), emergency departments (EDs) and inpatient wards in Queensland, New South Wales and South Australia, for children under 15 years receiving care in 2012–13. Our purpose was to estimate the percentage adherent for each indicator.

### Results

The medical records of 629 children with head injury were examined. Fifty-one percent of children were under 5 years old, with more males (61%) than females. Thirty-eight indicators were assessed. Avoidance of nasotracheal airways (100%; 95% CI: 99.4–100) or nasogastric tubes (99.7%; 95% CI: 98.5–100) for children with a head injury had the highest

Four ethics committees approved this data extraction without consent and would need to approve the release of data collected by the project, to ensure protection of both healthcare providers and individual patients. Most of the data used for calculation of weights is owned by third parties, and its release will be subject to third party approvals from: three state health departments (populations by health district, total ED presentations and inpatient admission numbers by hospital, percentage of ED admissions by condition), the Australian Government Department of Human Services (total number of consultations with children by General Practitioners and community paediatricians), the Australian Paediatric Research Network (percentage of consultations for each condition by community paediatricians) and the Bettering the Evaluation and Care of Health Program (percentage of consultations by condition for General Practice). Requests for access to data should be made in the first instance to the CTK Data Manager gaston.arnolda@mq.edu.au. Approval of all bodies from whom permissions are required will be needed.

**Funding:** The research was funded as an Australian National Health and Medical Research partnership grant (APP1065898) (JB), with contributions by the National Health and Medical Research Council, Bupa Health Foundation, Sydney Children's Hospital Network, New South Wales Kids and Families, Children's Health Queensland, and the South Australian Department of Health (SA Health). No funding bodies had any role in study design, data collection and analysis, decision to publish, or preparation of the manuscript. The authors have declared that no competing interests exist.

**Competing interests:** The authors have declared that no competing interests exist.

adherence. Indicators relating to primary and secondary assessment of head injuries were mostly adhered to. However, adherence to other indicators was poor (e.g., *documentation of the past history of children (e.g., presence or absence of seizures) before the injury*; 29.9% (95% CI: 24.5–35.7)), and for others was difficult to estimate with confidence due to small sample sizes (e.g., *Children with a head injury who were intubated had PaO$_2$ above 80mm Hg*; 56.0% (95% CI: 28.6–80.9)). Indicators guiding clinical decision making regarding the need for CT scan had insufficient data to justify reporting.

## Conclusion

This study highlights that management of head injury in children mostly follows guidelines, but also flags some specific areas of inconsistency. Individual sites are encouraged to use these results to guide investigation of local practices and inform quality improvement endeavours.

## Introduction

Head injuries, in which individuals suffer trauma to the head as a result of a collision or fall, is a common occurrence in children. Head injuries are classified as mild, moderate or severe based on Glasgow Coma Scale (GCS) ratings in routine clinical care. Traumatic head injury is the leading cause of trauma-related death and disability in both children and adults across the world and is linked to around half of all trauma-related deaths.[1, 2] Severe head injury can result in physical and intellectual disability, complications (such as epilepsy), and behavioural issues which can lead to unemployment and loss of independence, requiring lifelong care.[3] The total lifetime costs of moderate or severe traumatic brain injury (in 2008) to the Australian community were estimated to be AU$8.6 billion.[3]

Mild head injuries are common emergency and primary health presentations, comprising up to 90% of all head injuries presenting to ED.[4] In Australia and New Zealand, it is estimated that by the age of 15 years, 20% of children would have experienced a mild head injury. [4] Hospitalisations related to head injury occur more commonly in males than females, and more often in young children (mostly under three years).[5, 6]

Guidelines for management of head injury in children have been developed by several peak bodies including the National Institute of Health and Care Excellence (NICE)[7] and the American Academy of Paediatrics.[8] Clinical practice guidelines (CPGs) in this context seek to guide classification of mild, moderate and severe head injuries, and to identify the rare instances where an apparently mild head injury can mask a significant intracranial lesion. At the same time, CPGs try to limit overtreatment and distress for the majority of children whose injuries are benign. CPGs therefore identify specific signs and symptoms relevant to head injury to be documented (e.g., pupil size, breathing function), recommend tests to be undertaken (e.g., GCS, blood glucose), and outline appropriate management (e.g., medications for seizures, periods of observation for mild injury). Guidelines also define indications for Computed Tomography (CT) scanning, addressing concerns relating to unnecessary risks associated with radiation exposure in children[9] and unsustainable costs to health services.[8] Reasons for overuse of CT imaging in children are complex but include seeking to exclude serious injury, and providing a level of reassurance to clinicians, patients and carers.[10]

In addition to these specific guidelines for head injury presentations, CPGs also address more generic clinical examination of children and documentation of past history, co-morbidities and medications. Together these guidelines define best practice for children with a head injury.

CareTrack Kids (CTK) assessed the care of Australian children aged 0–15 years, in 2012 and 2013, to determine the proportion that were in line with guideline-based recommendations for 17 common conditions.[11] Across the 17 conditions, guideline-adherent care was estimated as being provided at an average of 59.8% of the time (95% CI: 57.5–62.0), and 78.3% (95% CI: 75.1–81.2) for head injury overall. In this paper we analyse and discuss the CareTrack Kids results for head injury, at indicator level.

## Methods

The CTK methods have been described in detail elsewhere.[11–14] We describe some aspects specifically relevant to head injury, with a focus on indicator development.

### Development of indicators

The RAND-UCLA method of indicator development[15] was adopted, with modifications. [12] For the purposes of this study, a clinical indicator was defined as a measurable component of a standard or guideline, with explicit criteria for inclusion, exclusion, time frame and practice setting.

We searched for Australian and international CPGs relating to head injury in children.[12] Recommendations were extracted to create initial draft indicators. Two CPGs were found for head injury and 105 recommendations extracted. Recommendations did not proceed to indicators if: they were guiding statements only with no recommended actions; used auxiliary verbs such as "may", "consider" and "could" to indicate the recommendation's strength; there was a low likelihood of information being documented in the medical record; or, they were out of scope for our purposes (such as structure-level measures). After excluding such recommendations and merging similar recommendations, 25 candidate recommendations were submitted as guideline-based indicators to internal review.

Internal reviews were undertaken by three clinicians (two paediatricians and a General Practitioner [16]) involved in the CTK study using a three-round modified Delphi approach by email. Reviewers rated candidate guideline-based indicators for acceptability, feasibility, and impact, and excluded eight. Thus, 17 guideline-based indicators were passed to external review by five paediatricians external to the project, recruited via advertisements and communications in relevant medical colleges, and professional associations and networks. External reviewers undertook a three-round modified Delphi approach, conducted on a custom-designed Wiki site, rating recommendations with the same criteria as internal reviewers, and also using a 9-point Likert scale to score each recommendation as representative of appropriate care delivered to Australian children during 2012 and 2013.[12] No indicators were removed during external review.

The 17 final guideline-based indicators were re-formatted into 54 medical record audit indicator questions; all indicator questions are shown in **S1**. Selected additional details on indicator development can be found in **S2**.

### Sample size, sampling process and data collection

CTK targeted 400 medical records for head injury and 6,000 medical records for 16 other conditions. If any of the 6,400 targeted records contained an occasion of care for head injury, a separate assessment of adherence was made for each relevant indicator during each visit. Detail on the general sampling methods are provided in the report of top-level results.[11] Additional

details specific to head injury can be found in in **S2**. Briefly, we sampled three health care settings: hospital inpatients, Emergency Department (ED) presentations, and consultations with GPs in randomly selected health districts in Queensland, New South Wales and South Australia, for children aged ≤ 15 years receiving care in 2012 and 2013. For the broader CTK study, the recruitment rate was 92% for hospitals, and estimated to be 24% for GPs (**see** S1 File). Data were collected by nine experienced paediatric nurses, trained and evaluated over five days to assess eligibility for indicator assessment and adherence.

### Analysis

At indicator level, estimates of adherence were measured as the percentage of eligible indicators (i.e., indicators scored either 'Yes' or 'No') which were scored as 'Yes'. Adherence for some clinically related indicators were aggregated as bundles of care, some of which were sub-grouped as sub-bundles. For example, indicators HEAD05-HEAD15 all relate to appropriate assessment of children who presented with a moderate to severe head injury; all eleven of these indicators would have to be scored 'Yes' for the bundle to be scored as adhering to the CPG. This bundle was also assessed as two sub-bundles, HEAD05-HEAD10 dealing with the primary assessment, and HEAD11-15 dealing with secondary assessment. When assessing bundles or sub-bundles, a visit was only included if there were responses for all component indicators.

Sampling weights were constructed as specified in S2, to control for oversampling of some states and care settings. The weighted data were analysed in SAS/STAT version 9.4 (SAS Institute Inc, North Carolina, USA), using the SURVEYFREQ procedure. Variance was estimated by Taylor series linearization. State and healthcare setting were specified as strata, where applicable, and the primary sampling unit (health district) was specified as the clustering unit. Exact 95% CIs were generated using the modified Clopper-Pearson method, except when the point estimate was 0% or 100%, where the unmodified Clopper-Pearson method was used.[17] In both indicator and bundle/sub-bundle reports, results were suppressed if there were <25 eligible visits. As they are independent of each other, direct comparisons between GP and inpatient/ED settings were performed by chi-square tests using weighted data; ED and inpatient records are related, so no comparisons were made. Statistical significance was computed for the difference between healthcare settings using the F-test approximation of the Rao-Scott chi-square test, which adjusts for the design effect.

### Ethical considerations

Primary ethics approval was from relevant bodies including hospital networks (HREC/14/SCHN/113; HREC/14/QRCH/91; HREC/14/WCHN/68) and the Royal Australian College of General Practitioners (NREEC 14–008), and site-specific approvals from 34 hospital sites. Australian Human Research Ethics Committees can waive requirements for patient consent for external access to medical records if the study entails minimal risk to providers and patients; [14] all relevant bodies provided this approval. Ethics approvals included the ability to report data by healthcare setting type when focusing on individual conditions. Participants were protected from litigation by gaining statutory immunity for CTK as a quality assurance activity, from the Federal Minister for Health under Part VC of the Health Insurance Act 1973 (Commonwealth of Australia).

### Results

Details of the 629 children with one or more eligible assessments of CPG adherence for head injury are provided in Table 1. Over half the children in the CTK sample were under five years

**Table 1. Characteristics of the 629 children, 2012–2013.**

| Characteristic | Children in the CTK Study |
|---|---|
| Age*—no. (%) | |
| < 1 year | 67 (10.7) |
| 1–2 years | 175 (27.8) |
| 3–4 years | 83 (13.2) |
| 5–11 years | 198 (31.5) |
| 12–15 years | 106 (16.9) |
| Male—no. (%) | 385 (61.2) |

* The child's age was calculated as the age at visit where there was only one, or the midpoint of the child's age at her first and last head injury visit.

of age, with more males (61%) than females. Each child was eligible for 1–4 head injury visits (median = 1). Of 40,662 possible indicator assessments, 5,675 (14.0%) were automatically filtered out by age or healthcare setting restrictions, and 24,709 (60.8%) were designated as not applicable by surveyors or otherwise ineligible (e.g., date of visit out of range). The field team conducted 10,278 eligible indicator assessments grouped into 746 visits, at a median of 13 indicators per visit. Eligible head injury visits were conducted in 53 GP practices, 34 hospital EDs and 26 hospital inpatient service providers—assessments by state and healthcare provider type are seen in Fig 1.

## Guideline adherence

The estimated adherence for each indicator is shown in Table 2. Adherence was not reported for 16 of the 54 indicators, because they had <25 visits. For the 38 reported indicators, adherence ranged from 29.9% for indicator HEAD20 ("*Children who presented with a head injury had their history documented which included the presence/absence of seizures*"; 95% CI: 24.5–35.7) to 100% for HEAD46 ("*Children who presented with a head injury were not intubated via a nasotracheal airway*"; 95% CI: 99.4–100); the latter representing avoidance of a non-recommended treatment. The interquartile range for adherence in the 38 indicators reported was 64.4% to 89.7%. Other higher adherent single indicators were for avoidance of nasogastric tubes (HEAD47; 99.7%; 95% CI: 98.5–100.0), primary survey and assessment of pupil size and reactivity (HEAD08; 98.0%; 95% CI: 90.4–99.9), documentation of the mechanism of injury (HEAD17; 97.2%; 95% CI: 94.2–98.9), and documentation of loss of consciousness (HEAD19; 92.3%; 95% CI: 86.1–96.3).

There was insufficient data to estimate appropriate assignment of Triage 1 classification, but estimates were above 80% for all other categories. Estimated adherence was 90.2% for classification of Triage 2 patients (95% CI: 65.2–99.2; HEAD02), 82.9% for Triage 3 (95% CI: 69.5–92.1; HEAD03) and 92.5% for Triage 4/5 (95% CI: 80.3–98.3; HEAD04).

The assessed adherence of two bundles is shown in Table 3. Both bundles were assessed in all three healthcare settings, and Bundle A was broken into two sub-bundles. Bundle A assessed the documentation of 11 assessments and found an overall 47.1% adherence (95% CI: 27.1–67.7); this bundle was made up of two sub-bundles, one comprising six indicators which related to primary assessment (54.1%; 95% CI: 32.8–74.4), and the other five relating to secondary assessment (58.2%; 95% CI: 40.5–74.5). Individual indicators with estimated adherence under 80% in Bundle A related to: primary survey for airway assessment with cervical spine immobilization (HEAD05; 75.1%; 95% CI: 57.1–88.4) and blood glucose assessment (HEAD10; 58.5%; 95% CI: 40.1–75.3); and the secondary survey for possible scalp injuries (HEAD11; 76.4; 95% CI: 63.6–86.5), base of

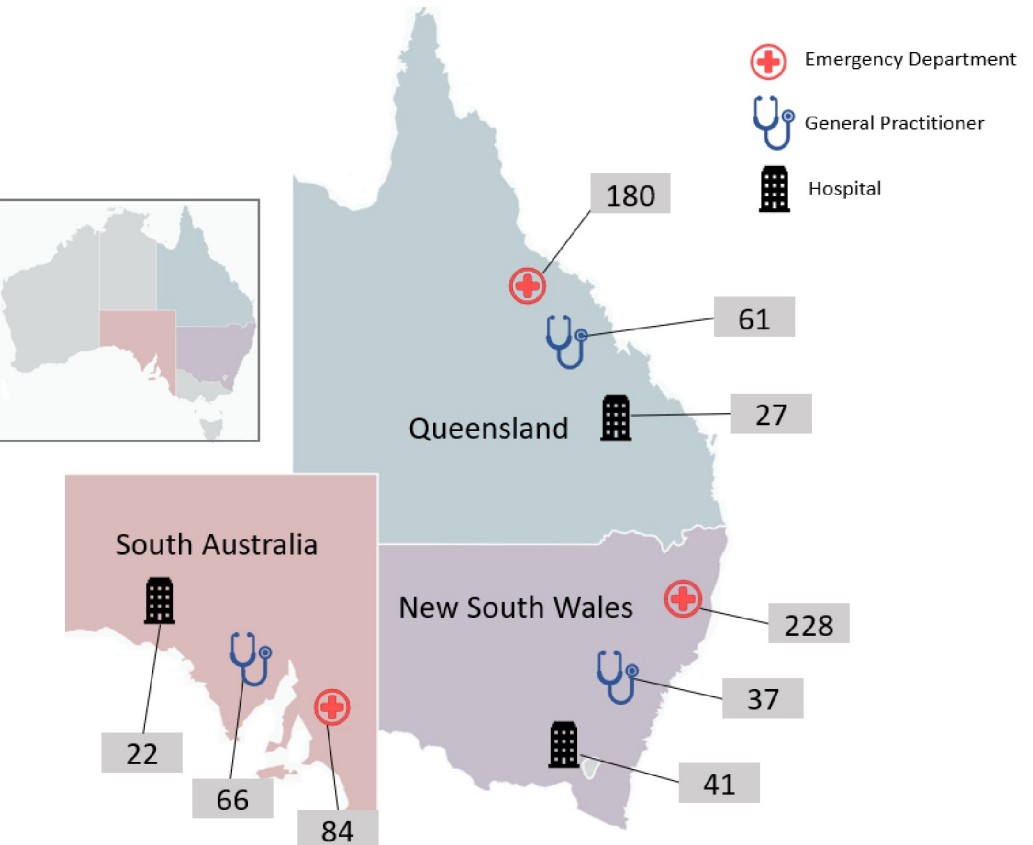

**Fig 1. Head injury assessments by state and healthcare provider type.** Total number of visits to emergency departments = 492; total number of admissions to hospital = 90; and total number of visits to general practitioners = 164. Total number of head injury assessments in: New South Wales = 306; Queensland = 268; and South Australia = 172. Total number of visits assessed for care of head injury in sampling frame = 746.

skull fractures (HEAD12; 72.6%; 95% CI: 56.7–85.2), CSF leaks, or haemo-tympanum (HEAD13; 64.4%; 95% CI: 48.5–78.4).

Bundle B covered nine indicators relating to the documentation of history and found 18.3% adherence. The component indicator with the lowest adherence was documentation of the presence/absence of seizures prior to the injury (HEAD20; 29.9%). Other indicators in Bundle B with estimated adherence under 80% were documentation of the time of injury (HEAD16; 76.2%; 95% CI: 67.3–83.8), nausea or vomiting (HEAD22; 78.5%; 95% CI: 71.9–84.2) and clinical course prior to consultation (HEAD23; 63.1%; 95% CI: 55.9–70.0%). Also concerned with history, but not included in Bundle B, HEAD25 addresses documentation of comorbidities that predispose to intracranial injury; estimated adherence was 46.4% (95% CI: 31.7–61.6).

Outside of the Bundles, a large number of indicators had either insufficient data to justify publishing estimated adherence, or wide confidence intervals as they were assessed in fewer than 50 visits. For example, three indicators (HEAD48-50) relating to management of intubated patients showed lower adherence but with wide 95% CIs: 37.4% (95% CI: 14.2–66.0) for end tidal $CO_2$ monitoring; 56.0% (95% CI: 28.6–80.9) for maintaining $PaO_2$ above 80mm Hg; and 45.6% (95% CI: 19.5–73.6) for maintaining $PaCO_2$ levels between 35-40mm Hg. Indicators with sufficient data, not already discussed above, include the receipt of cervical spine precautions for selected groups of head injury patients (HEAD51; 77.8%; 95% CI: 63.1–88.8), and

**Table 2. Adherence by clinical indicator, 2012–2013.**

| Indicator ID | Indicator Description | No. of Children | No. of Visits | Proportion Adherent % (95% CI) |
|---|---|---|---|---|
| HEAD01 | Children who presented with a head injury and any of the following: * unconscious/responding only to pain OR * fitting OR * signs of cardiovascular compromise were categorised as a Triage 1 patient. | 12 | 12 | Insufficient data |
| HEAD02 | Children who presented with a head injury and any of the following: * abnormal drowsiness/ responding only to voice OR * loss of consciousness of more than 5 minutes OR * focal signs OR * severe pain or headache OR * high risk mechanism were categorised as a Triage 2 patient. | 53 | 60 | 90.2 (65.2, 99.2) |
| HEAD03 | Children who presented with a head injury and any of the following: * alert but altered behaviour OR * loss of consciousness less than 5 minutes OR * moderate pain or headache OR * moderate risk mechanism OR * significant neurological, developmental or bleeding comorbidities OR * less than one year of age OR * possible inflicted head injury, otherwise well were categorised as a Triage 3 patient. | 272 | 308 | 82.9 (69.5, 92.1) |
| HEAD04 | Children aged ≥ 12 months who presented with an acute head injury and ONLY the following features: * low impact mechanism AND * NO neurological signs or symptoms AND * NO comorbidities or concerns regarding inflicted head injury were categorised as a Triage 4 or 5 patient. | 133 | 146 | 92.5 (80.3, 98.3) |
| HEAD05 | Children who presented with a moderate to severe head injury (GCS 3–13) received a primary survey and assessment of their airway (with cervical spine immobilisation). | 52 | 57 | 75.1 (57.1, 88.4) |
| HEAD06 | Children who presented with a moderate to severe head injury (GCS 3–13) received a primary survey and assessment of their breathing function. | 52 | 57 | 89.7 (66.9, 98.8) |
| HEAD07 | Children who presented with a moderate to severe head injury (GCS 3–13) received a primary survey and assessment of their circulation. | 53 | 58 | 89.0 (66.9, 98.4) |
| HEAD08 | Children who presented with a moderate to severe head injury (GCS 3–13) received a primary survey and assessment of their pupil size and reaction to light. | 53 | 58 | 98.0 (90.4, 99.9) |
| HEAD09 | Children who presented with a moderate to severe head injury (GCS 3–13) received a primary survey and assessment of their GCS or AVPU. | 53 | 58 | 95.5 (86.6, 99.2) |
| HEAD10 | Children who presented with a moderate to severe head injury (GCS 3–13) received a primary survey and assessment of their blood glucose. | 53 | 58 | 58.5 (40.1, 75.3) |
| HEAD11 | Children who presented with a moderate to severe head injury (GCS 3–13) received a secondary survey which included palpation for bogginess, swelling or bruising of the scalp. | 54 | 61 | 76.4 (63.6, 86.5) |
| HEAD12 | Children who presented with a moderate to severe head injury (GCS 3–13) received a secondary survey which included looking for signs of base of skull fracture such as Battle's sign (bruising over mastoid), 'raccoon' eyes or blood behind the ear drum. | 54 | 60 | 72.6 (56.7, 85.2) |
| HEAD13 | Children who presented with a moderate to severe head injury (GCS 3–13) received a secondary survey which included examination for haemo-tympanum or signs of CSF leak from ears or nose. | 54 | 61 | 64.4 (48.5, 78.4) |
| HEAD14 | Children who presented with a moderate to severe head injury (GCS 3–13) received a secondary survey which included an examination for facial (e.g. nose, mouth, ears) deformities, swelling, bleeding, lacerations, tenderness. | 54 | 61 | 93.8 (84.6, 98.4) |
| HEAD15 | Children who presented with a moderate to severe head injury (GCS 3–13) received a secondary survey which included examination for cervical spine deformity, tenderness, muscle spasm, crepitus, motor function, reflexes and lateralising signs. | 54 | 61 | 89.7 (76.0, 97.0) |
| HEAD16 | Children who presented with a head injury had their history documented which included the time of injury. | 624 | 737 | 76.2 (67.3, 83.8) |
| HEAD17 | Children who presented with a head injury had their history documented which included mechanism of injury. | 627 | 742 | 97.2 (94.2, 98.9) |
| HEAD18 | Children who presented with a head injury had their history documented which included a recall of events. | 585 | 694 | 85.6 (79.2, 90.7) |
| HEAD19 | Children who presented with a head injury had their history documented which included whether there was loss or impairment of consciousness (and duration). | 626 | 741 | 92.3 (86.1, 96.3) |
| HEAD20 | Children who presented with a head injury had their history documented which included the presence/absence of seizures. | 625 | 739 | 29.9 (24.5, 35.7) |
| HEAD21 | Children who presented with a head injury had their history documented which included their behaviour and activity since the time of injury. | 627 | 742 | 88.6 (80.1, 94.4) |
| HEAD22 | Children who presented with a head injury had their history documented which included whether they had any nausea or vomiting. | 627 | 742 | 78.5 (71.9, 84.2) |

(*Continued*)

**Table 2.** (Continued)

| Indicator ID | Indicator Description | No. of Children | No. of Visits | Proportion Adherent % (95% CI) |
|---|---|---|---|---|
| HEAD23 | Children who presented with a head injury had their history documented which included their clinical course prior to consultation, e.g. stable, deteriorating, improving. | 627 | 742 | 63.1 (55.9, 70.0) |
| HEAD24 | Children who presented with a head injury had their history documented which included any other injuries sustained. | 627 | 742 | 88.3 (82.1, 93.0) |
| HEAD25 | Children who presented with a head injury had their history documented which included comorbidities that predispose to intracranial injury (intra-cerebral shunt, AV malformation, bleeding disorders (including vitamin K deficiency). | 381 | 461 | 46.4 (31.7, 61.6) |
| HEAD26 | Children who presented to the ED with a head injury and any of the following: * GCS persistently less than or equal to 8 OR * loss of protective laryngeal reflexes OR * abnormal breathing pattern or hypoventilation OR * oxygen saturation less than or equal to $SpO_2$ 95% or a $PaO_2$ less than 80 mmHg on maximal facial oxygen OR * $PaCO_2$ less than 30 mmHg or $PaCO_2$ greater than 44 mmHg were classified as severe and were intubated and ventilated. | 11 | 11 | Insufficient data |
| HEAD27 | Children with a severe head injury (GCS 3–8) received immobilisation of their cervical spine. | 17 | 19 | Insufficient data |
| HEAD28 | Children with a severe head injury (GCS 3–8) who had completed their fluid resuscitation, were nursed 20–30 degrees head up. | 11 | 11 | Insufficient data |
| HEAD29 | Children with a severe head injury (GCS 3–8) received continuous cardio-respiratory (respiratory rate, pulse) and oxygen saturation monitoring. | 16 | 18 | Insufficient data |
| HEAD30 | Children with a severe head injury (GCS 3–8) had their BP measured every 15–30 minutes. | 16 | 18 | Insufficient data |
| HEAD31 | Children with a severe head injury (GCS 3–8) who were not intubated, had their GCS recorded every 15–30 minutes. | 8 | 8 | Insufficient data |
| HEAD32 | Children with a severe head injury (GCS 3–8) received an urgent CT of the head. | 15 | 17 | Insufficient data |
| HEAD33 | Children with a severe head injury (GCS 3–8) received an urgent C-Spine CT. | 15 | 17 | Insufficient data |
| HEAD34 | Children with a severe head injury (GCS 3–8) received a consultation with ICU and neurosurgical specialists. | 15 | 16 | Insufficient data |
| HEAD35 | Children who presented with moderate head injury (GCS 9–13) without neurological deterioration had their GCS observed in hospital at least half-hourly for a minimum of four hours. | 26 | 27 | 72.4 (45.1, 91.2) |
| HEAD36 | Children who presented with moderate head injury (GCS 9–13) without neurological deterioration had their pulse rate observed in hospital at least half-hourly for a minimum of four hours. | 30 | 31 | 80.0 (56.1, 94.3) |
| HEAD37 | Children who presented with moderate head injury (GCS 9–13) without neurological deterioration had their respiratory rate observed in hospital at least half-hourly for a minimum of four hours. | 30 | 31 | 80.0 (56.1, 94.3) |
| HEAD38 | Children who presented with moderate head injury (GCS 9–13) without neurological deterioration had their blood pressure observed in hospital at least half-hourly for a minimum of four hours. | 30 | 31 | 74.9 (48.4, 92.3) |
| HEAD39 | Children who presented with moderate head injury (GCS 9–13) without neurological deterioration had their pupils assessed in hospital at least half-hourly for a minimum of four hours. | 30 | 31 | 71.6 (45.7, 90.0) |
| HEAD40 | Children who presented with moderate head injury (GCS 9–13) without neurological deterioration had their limb strength assessed in hospital at least half-hourly for a minimum of four hours. | 30 | 31 | 69.2 (42.8, 88.7) |
| HEAD41 | Children with a moderate/intermediate head injury (GCS 9–13) who experienced an acute deterioration including persistent vomiting (at 6 hours post injury) received a CT of the head. | 11 | 11 | Insufficient data |
| HEAD42 | Children with a moderate/intermediate head injury (GCS 9–13) who experienced an acute deterioration including persistent headache (at 6 hours post injury) received a CT of the head. | 4 | 4 | Insufficient data |
| HEAD43 | Children with a moderate/intermediate head injury (GCS 9–13) who experienced an acute deterioration including persistent irritability (at 6 hours post injury) received a CT of the head. | 5 | 5 | Insufficient data |
| HEAD44 | Children with a moderate/intermediate head injury (GCS 9–13) who experienced an acute deterioration including persistent abnormal behaviour/neurological abnormality (at 6 hours post injury) received a CT of the head. | 10 | 10 | Insufficient data |
| HEAD45 | Children with a moderate/intermediate head injury (GCS 9–13) who experienced an acute deterioration including persistent unsteady gait (at 6 hours post injury) received a CT of the head. | 1 | 1 | Insufficient data |
| HEAD46 | Children who presented with a head injury were not intubated via a nasotracheal airway. | 462 | 574 | 100 (99.4, 100) |
| HEAD47 | Children who presented with a head injury did not receive a nasogastric tube. | 464 | 578 | 99.7 (98.5, 100) |
| HEAD48 | Children with a head injury who were intubated had end tidal $CO_2$ monitoring. | 30 | 35 | 37.4 (14.2, 66.0) |
| HEAD49 | Children with a head injury who were intubated had $PaO_2$ greater than 80 mmHg ($SaO_2$ greater than 95%). | 23 | 27 | 56.0 (28.6, 80.9) |

*(Continued)*

**Table 2.** (Continued)

| Indicator ID | Indicator Description | No. of Children | No. of Visits | Proportion Adherent % (95% CI) |
|---|---|---|---|---|
| HEAD50 | Children with a head injury who were intubated had $PaCO_2$ between 35–40 mmHg. | 22 | 26 | 45.6 (19.5, 73.6) |
| HEAD51 | Children who presented with a head injury and any of the following: * GCS less than 15 OR * posterior bony neck pain or tenderness OR * focal deficit at any time since injury OR * paraesthesia in the extremities OR * distracting injury OR * intoxication received cervical spine precautions. | 65 | 75 | 77.8 (63.1, 88.8) |
| HEAD52 | Children who presented with head injury who were seizing, were immediately administered: * midazolam (0.15 mg/kg bolus IV), OR * diazepam (0.25 mg/kg bolus IV) OR * midazolam 0.15 mg/kg IM, 0.5 mg/kg IN or 0.5 mg/kg buccal. | 6 | 7 | Insufficient data |
| HEAD53 | Children who presented with head injury and received sedation and/or opioid analgesia had their GCS recorded every 15 minutes until their GCS returned to the pre-sedation level. | 30 | 33 | 61.5 (43.0, 77.9) |
| HEAD54 | Children with a minor/mild head injury (GCS 14–15) whose parents were provided with information on when to return to the ED if deterioration occurs, were discharged from the ED without a period of observation. | 294 | 317 | 58.2 (45.7, 70.0) |

GCS = Glasgow Coma Scale; AVPU = Alert/Pain/Voice/Unresponsive; CSF = Cerebrospinal Fluid; AV = arteriovenous; $CO_2$ = Carbon dioxide; $PaO_2$ = Partial pressure of oxygen; $PaCO_2$ = Partial pressure of carbon dioxide; $SaO_2$ = Arterial oxygen saturation; IV = Intravenous; IM = Intra-muscular; IN = Intra-nasal; ED = Emergency Department.

prompt discharge of ED patients for mild cases, with instructions to return if there is deterioration (HEAD54; 58.2%; 95% CI: 45.7–70.0).

Overall estimates for each health care setting (i.e., GP, ED or inpatient) available in each context (i.e., metropolitan or regional geographical location, or tertiary paediatric hospital) is shown in Table 4. Estimated adherence across all indicators was similar for each of GP, ED and inpatient settings regardless of setting context, but estimated adherence in GP settings was over 20 percentage points lower than for either ED or inpatient settings in both metropolitan

**Table 3. Adherence by bundle of care, 2012–2013.**

| Bundle ID | Bundle Description | Indicator IDs* | Healthcare Setting | No. of Children | No. of Visits | Proportion Adherent, % (95% CI) |
|---|---|---|---|---|---|---|
| A | Children who presented with a moderate to severe head injury received appropriate survey and assessment. | 05–15 | GP | 3 | 3 | Insufficient data |
| | | | ED | 42 | 43 | 43.4 (16.2, 73.9) |
| | | | Inpatient | 10 | 10 | Insufficient data |
| | | | Overall | 51 | 56 | 47.1 (27.1, 67.7) |
| A.1 | Children who presented with a moderate to severe head injury received appropriate primary survey and assessment. | 05–10 | GP | 3 | 3 | Insufficient data |
| | | | ED | 42 | 43 | 49.8 (24.2, 75.5) |
| | | | Inpatient | 10 | 10 | Insufficient data |
| | | | Overall | 51 | 56 | 54.1 (32.8, 74.4) |
| A.2 | Children who presented with a moderate to severe head injury received appropriate secondary survey. | 11–15 | GP | 3 | 3 | Insufficient data |
| | | | ED | 44 | 45 | 54.4 (28.4, 78.6) |
| | | | Inpatient | 12 | 12 | Insufficient data |
| | | | Overall | 54 | 60 | 58.2 (40.5, 74.5) |
| B | Children who presented with a head injury had their history documented. | 16–24 | GP | 153 | 155 | 2.6 (0.1, 11.5) |
| | | | ED | 413 | 449 | 24.4 (16.3, 34.0) |
| | | | Inpatient | 82 | 82 | 32.6 (16.6, 52.2) |
| | | | Overall | 579 | 686 | 18.3 (13.8, 23.7) |

GP, General practice; ED, Emergency Department.

* In Table 2, the indicator ID was preceded by 'HEAD'.

**Table 4. Adherence of care by geographical regions and tertiary hospitals, and healthcare setting.**

| Geographical Regions and Tertiary Hospitals* | Healthcare Setting | No. of Children | No. of Visits | No. of Indicators Assessed | Proportion Adherent, % (95% CI) |
|---|---|---|---|---|---|
| Metropolitan | GP | 115 | 117 | 1136 | 62.1 (51.7, 71.6) |
| | ED | 169 | 178 | 2794 | 85.1 (81.7, 88.1) |
| | Inpatient | 29 | 29 | 434 | 84.9 (71.5, 93.6) |
| Regional | GP | 47 | 47 | 449 | 58.6 (53.3, 63.7) |
| | ED | 218 | 243 | 3556 | 82.0 (79.0, 84.7) |
| | Inpatient | 37 | 37 | 470 | 79.8 (65.7, 90.0) |
| Tertiary paediatric hospitals | ED | 66 | 71 | 1025 | 82.8 (65.7, 93.7) |
| | Inpatient | 24 | 24 | 414 | 86.6 (70.6, 95.8) |

(p<0.0001 in both settings) and regional contexts (both p<0.0001). Overall adherence was highest for inpatient tertiary paediatric hospitals (86.6%, 95% CI: 70.6, 95.8).

## Discussion

This study estimated the adherence to care guidelines for children aged 0–15 years who presented with a head injury to GP practices, EDs and inpatient services. Of the 54 indicators assessed, a total of 38 yielded sufficient data to analyse individually. Indicators that were inappropriate for GP practice settings (e.g., HEAD01-04 regarding triage categories) were only collected in EDs and/or inpatient settings. Overall, the interquartile range for guideline adherence for the 38 indicators was found to be 64.4% to 89.7%.

Indicators with the highest adherence related to avoidance of nasotracheal airways (HEAD46; 100%) and nasogastric tubes (HEAD47; 99.7%). Nasotracheal and nasogastric intubation is contra-indicated in head injury due to the rare but devastating complication of tube misplacement into the brain.[18] This has been reported in head injuries involving facial fractures, especially undiagnosed fractures of the basal skull, sphenoid sinus, or cribriform plate.[19] Next highest adherence for single indicators were for primary survey and assessment of pupil size and reactivity (HEAD08; 98.0%), documentation of the mechanism of injury (HEAD17; 97.2%), and documentation of loss of consciousness (HEAD19; 92.3%).

Incomplete undertaking or recording of primary and secondary assessments of head injury flag a risk for children provisionally diagnosed with a mild head injury, whose condition later deteriorates, as baseline data will not be available. Moreover, a complete picture of pre-existing conditions and co-morbidities are needed to make appropriate clinical decisions such as CT imaging. There was a contrast between indicators with high adherence and others with low adherence within these two bundles when the individual indicators were considered.

Less than half of the children with moderate to severe head injury had full documentation of primary and secondary examination findings (Bundle A). Lowest adherence within this Bundle, at 58.5%, was the assessment of blood glucose levels (HEAD10). Blood glucose level on presentation is important for two main reasons in a child with a head injury. Firstly, it may reveal hypoglycaemia as a differential diagnosis to consider for lowered (or subsequently changing) levels of consciousness. Blood glucose level on primary assessment has also been suggested as a sensitive indicator of brain injury severity with early hyperglycaemia found to be predictive of a poorer prognosis.[20–22]

Other indicators in the primary and secondary assessment bundles that showed relatively low levels of adherence were failure to document the past history of children including presence or absence of seizures (HEAD20; 29.9%), and failure to document co-morbidities that pre-dispose children to intracranial injury such as intra-cerebral shunts, AV malformation,

or bleeding disorders (HEAD25; 46.4%). These are important complicating factors to check and document so that accurate baseline assessment can be measured, and appropriate management commenced. In relation to co-morbidities, even minor head injuries (e.g., a fall while playing) in children with a bleeding disorder such as haemophilia A, B, or von Willebrand's disease may lead to intracranial bleeding, and change the threshold criteria for which CT imaging is warranted.[23] Previous intracranial surgery or known malformations also heighten the risk of serious sequelae. It is noted that surveyors of the medical records were asked to use their clinical judgement when assessing specific indicator questions. If a child had a documented comment such as "normally well", then this assumes that the clinician checked for a history of seizures or co-morbidities, and should have marked it as adherent.

A point of interest is that there is a contrast in adherence between the different recommended factors within the assessment, e.g., the contrast between assessment of pupil size and reactivity (98.0%) and documentation of pre-existent seizures: HEAD20 (29.9%). This may be a reflection of documentation habits where only some factors are noted, and others not (e.g., the absence of seizure history is not documented, only its presence) rather than actual clinical reasoning. It does suggest that clinicians could improve the quality of their assessments by documenting these extra criteria and they are encouraged to consider adjusting their routine documentation to include them.

The cervical spine is at risk of injury in children who suffer a head trauma, meaning cervical immobilisation is indicated in some cases to prevent exacerbation of any underlying injury. Adherence to HEAD05 which refers to cervical immobilization in children with moderate to severe head injury (GCS 3–8) was estimated at 75.1%, suggesting room for improvement. Possible explanations for this lower than expected figure may be that collars are difficult to fit in young children, patients who are agitated will not tolerate a collar, the mechanism of injury may not have placed the spine at risk, or it was done but not documented.

An important aim of primary management of a child with a head injury is to prevent secondary damage to the brain by ensuring optimal oxygenation and ventilation. Three important indicators (HEAD48-50) relating to management of intubated patients showed lower adherence but with wide 95% CIs: 37.4% (95% CI: 14.2–66.0) for end tidal $CO_2$ monitoring; 56.0% (95% CI: 28.6–80.9) for maintaining $PaO_2$ above 80mm Hg; and 45.6% (95% CI: 19.5–73.6) for maintaining $PaCO_2$ levels between 35-40mm Hg. The CI width reflects the low number of cases audited (n = 26–35) but suggests that these indicators should be checked locally as part of routine quality assurance activities.

The indicator HEAD54 ("Children with a mild head injury (GCS 14–15) whose parents were provided with information on when to return to the ED if deterioration occurs, were discharged from the ED without a period of observation") was only documented 58.2% of the time (95% CI: 45.7–70.0). This indicator has two aspects; provision of information to the parents, and not keeping the child for a period of observation. In hindsight, this indicator may have more usefully been split into two as adherence rates do not tell us which aspect was not met. The low adherence may indicate a more conservative policy of keeping children in for a period of observation, which may not be strictly warranted by the GCS and assessment. Alternatively, it may indicate a lack of parental education. The importance of providing information to parents as the primary caregivers is a key safety issue to ensure prompt action in the case of deterioration. It is quite possible that this was a case of documentation failure rather than action failure for low risk presentations. Either way, these results highlight another area of patient safety that EDs may wish to target for improvement.

Estimated adherence was very similar across healthcare context for GP care (Metro and Regional), and for both ED and inpatient care (Metro, Regional, and Tertiary Hospitals). Within each healthcare context, however, estimated GP adherence (~60% in both contexts)

was over 20 percentage points lower than either ED or inpatient adherence (all ~80–85%) in both the Metro and Regional contexts. A possible explanation is that children with lower severity of head injury were over-represented in the GP setting and confidence that patients were at low risk for adverse sequelae may have led to under-documentation. Electronic documentation of medical assessment used in EDs may have structured fields that may be more likely to be completed than unstructured hand-written or free text electronic documentation.[24] This may have acted to increase the adherence of documentation in hospitals. Overall, adherence was high for other geographic and health care settings ranging from 79.8% (95% CI: 65.7, 90.0) for regional hospital inpatient settings to 86.6% (95% CI: 70.6, 95.8) in the tertiary paediatric inpatient setting.

Issues around the nature of CPGs may have been influential. As discussed in the main results paper of the CTK program,[11] low adherence and practice variation may reflect the failure of CPGs to be effectively structured, presented and implemented.

## Strengths and limitations

The main strength of the study is that a multi-stage representative sample was taken, covering Australian states containing 60% of Australia's children, and as such is likely generalisable to much of the unsampled population. Rather than self-report, an audit of medical records allowed an assessment of real-world practice. The audits themselves were carried out by experienced paediatric Registered Nurses who had undergone extensive training for the task. The whole medical record was available to the auditors.

Due to the small sample size of children who met some inclusion criteria (e.g., indicators for children GCS 9–13), many of the indicators had insufficient data to analyse or report. For example, only twelve children presenting to an ED were admitted to an ED during the audit window with an injury indicating a Triage 1 status (HEAD01). Notably, our study had low numbers of children who met the criteria that indicated a need for CT scanning (HEAD32 and HEAD41-45). Identifying conditions for the use of CT scans is a main area for both guideline adherence research and guideline development in the paediatric head injury literature due to concerns around unnecessary radiation exposure in children[9] and unsustainable costs to health services.[8] The low numbers mean that our study cannot add to the literature in this particular aspect.

An acknowledged weakness of the study is the use of documentation to assess actual practice; i.e., assuming that the action was not done if it was not documented. We note that "false positives" are also a possibility; i.e., documentation without action.[25] All indicators were chosen as having a high expectation of being documented. Further, we argue that from a litigation, billing, insurance or auditing point of view, documentation is the accepted means of indicating action.[26] More importantly, documentation is a crucial issue for patient safety and quality of care. Accurate and complete documentation ensures relevant information is recorded and accessible for all treating team members to enable robust diagnosis and management decisions, prevent duplication of tests or procedures already performed, and appropriate context to assess ongoing treatment and any adverse sequelae.

## Implications

Three areas of improvement are suggested from these results. Firstly, there are important aspects of primary and secondary assessment of head injury in children that were not consistently documented (e.g., past history of seizures, examination for haemo-tympanum or CSF leaks) in EDs and general practices. Review of current practice against these indicators by individuals and local units may be useful in identifying specific issues that could be addressed

actively to improve the quality of care. Secondly, there were low point estimates but wide confidence intervals around some of the estimates for important indicators related to intubation of children with a head injury (e.g., HEAD50 $PaCO_2$ was between 35–40 mmHg; 45.6%; (95% CI: 19.5–73.6)) which suggest that leaders within individual sites should review their intubation practice with an aim of achieving consistently high quality monitoring. Thirdly, there may be a role for peak bodies to review CPGs around head injury in children for clarity, currency and usability in the clinical setting.

## Conclusion

Head injuries in children are a common but potentially devastating presentation across EDs and general practices. CPGs guide the management of these injuries, providing clear assessment criteria on which to base treatment decisions and a baseline to monitor the course of disease and the effectiveness of interventions. This study analysed adherence to agreed indicators for best practice in head injury management across three health settings in three states of Australia. Eligible assessments involved 629 children in 53 GP practices, 34 hospital EDs and 26 hospital inpatient service providers. Overall adherence was 78.3% (95% CI: 75.1–81.2).[11] Further analysis of results showed that avoidance of nasotracheal airways (100%) or nasogastric tubes (99.7%) had the highest adherence. Indicators relating to primary and secondary assessment of head injuries were generally well done with adherence for most above 85%. At the same time, adherence to some indicators was poor (e.g., HEAD20: *Failure to document the past history of children (e.g., presence or absence of seizures) before the injury*; 29.9% (95% CI: 24.5–35.7), and for others were uncertain because of small sample size (e.g., HEAD49 *Children with a head injury who were intubated had $PaO_2$ above 80mm Hg*; 56.0% (95% CI: 28.6–80.9). Results may be used to flag individual issues that may be sub-optimal at individual sites and guide quality improvement endeavours.

## Supporting information

**S1 Table. Characteristics, by clinical indicator, 2012–2013.** ID = Identifier; GP = General Practitioner; ED = Emergency Department; IP = Inpatient; GCS = Glasgow Coma Scale; AVPU = Alert/Pain/Voice/Unresponsive; CSF = Cerebrospinal Fluid; AV = arteriovenous; $CO_2$ = Carbon dioxide; $PaO_2$ = Partial pressure of oxygen; $PaCO_2$ = Partial pressure of carbon dioxide; $SaO_2$ = Arterial oxygen saturation; IV = Intravenous; IM = Intra-muscular; IN = Intranasal; ED = Emergency Department. # Strength of recommendation as reported in individual CPGs. CPGs used a variety of classification schemes for allocating strength of recommendation in Grades (with A indicating the strongest recommendation in all classification schemes). If Grades were not specified in the CPG, or a Level of Evidence category, the term "Consensus-based recommendation" was assigned. * The type of quality of care assessed was classified as underuse or overuse: underuse refers to actions which are recommended, but not undertaken; overuse refers to actions which are not indicated or contraindicated in the context of the indicator's inclusion criteria.
(DOCX)

**S1 File. Additional details relating to study methods.**
(DOCX)

## Acknowledgments

We acknowledge the CareTrack Kids investigative team: Adam Jaffe, MD; Les White, DSc; Christopher T. Cowell, MBBS; Mark F. Harris, MD; William B. Runciman, MD; Andrew R.

Hallahan, MBBS; Gavin Wheaton, MBBS; Helena M. Williams, MBBS; Elisabeth Murphy, MPaeds; Shanthi Ramanathan, PhD; Tamara D. Hooper, BSc; Natalie Szabo, MA; John G. Wakefield, MPH; Clifford F. Hughes, DSc; Annette Schmiede, BEc; Chris Dalton, MPH; Joanna Holt, MHP; Liam Donaldson, MD; Ed Kelley, PhD; Richard Lilford, DSc; Peter Lachman, MD; Stephen Muething, MD.

## Author Contributions

**Conceptualization:** Janet C. Long, Peter D. Hibbert, Jeffrey Braithwaite.

**Data curation:** Gaston Arnolda, Hsuen P. Ting, Charlotte J. Molloy, Louise K. Wiles.

**Formal analysis:** Gaston Arnolda, Hsuen P. Ting, Charlotte J. Molloy, Peter D. Hibbert, Louise K. Wiles.

**Funding acquisition:** Jeffrey Braithwaite.

**Investigation:** Janet C. Long, Peter D. Hibbert, Jeffrey Braithwaite.

**Methodology:** Sarah Dalton, Gaston Arnolda, Hsuen P. Ting, Charlotte J. Molloy, Peter D. Hibbert, Louise K. Wiles.

**Project administration:** Janet C. Long, Meagan Warwick, Jeffrey Braithwaite.

**Supervision:** Janet C. Long, Jeffrey Braithwaite.

**Validation:** Sarah Dalton, Gaston Arnolda, Hsuen P. Ting, Charlotte J. Molloy, Louise K. Wiles, Simon Craig.

**Visualization:** Meagan Warwick.

**Writing – original draft:** Janet C. Long, Sarah Dalton, Gaston Arnolda, Hsuen P. Ting, Charlotte J. Molloy, Peter D. Hibbert, Louise K. Wiles, Simon Craig, Meagan Warwick, Kate Churruca, Louise A. Ellis, Jeffrey Braithwaite.

**Writing – review & editing:** Janet C. Long, Sarah Dalton, Gaston Arnolda, Hsuen P. Ting, Charlotte J. Molloy, Peter D. Hibbert, Louise K. Wiles, Simon Craig, Meagan Warwick, Kate Churruca, Louise A. Ellis, Jeffrey Braithwaite.

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
