## [Decision Letter · Decision Letter 0]

28 Oct 2019

PONE-D-19-26245

Guideline Adherence in the Management of Head Injury in Australian Children: A Population-Based Sample Survey

PLOS ONE

Dear Professor Braithwaite,

Thank you for submitting your manuscript to PLOS ONE. After careful consideration, we feel that it has merit but does not fully meet PLOS ONE’s publication criteria as it currently stands. Therefore, we invite you to submit a revised version of the manuscript that addresses the points raised during the review process.

We would appreciate receiving your revised manuscript by Dec 12 2019 11:59PM. To enhance the reproducibility of your results, we recommend that if applicable you deposit your laboratory protocols in protocols.io, where a protocol can be assigned its own identifier (DOI) such that it can be cited independently in the future. For instructions see: http://journals.plos.org/plosone/s/submission-guidelines#loc-laboratory-protocols

We look forward to receiving your revised manuscript.

Kind regards,

Itamar Ashkenazi

Academic Editor

PLOS ONE

Journal Requirements:

Reviewers' comments:

Reviewer's Responses to Questions

**Comments to the Author**

1. Is the manuscript technically sound, and do the data support the conclusions?

Reviewer #1: Partly

2. Has the statistical analysis been performed appropriately and rigorously? 

Reviewer #1: N/A

3. Have the authors made all data underlying the findings in their manuscript fully available?

Reviewer #1: Yes

4. Is the manuscript presented in an intelligible fashion and written in standard English?

Reviewer #1: Yes

5. Review Comments to the Author

Reviewer #1: Manuscript: Guideline adherence in the management of head injury in Australian Children: a population-based sample survey (PONE-D-19-26245)

Study type: survey of quality indicators

Authors' methodology and main findings: The authors compiled a list of quality indicators based upon guidelines and refined by a Delphi process. General practitioners' (27 of 114 approached), emergency department records, and hospitalized patient records of 629 children injured between 2012-2013 were evaluated. The authors found that adherence to avoidance of nasotracheal airways and nasogastric tubes was high, while documentation of past history and presence/absence of seizures was low.

Reviewer's comments: This article deals with an important issue of indicators in treatment of head injury. While the findings of this study are not generalizable, the method used may considered as generalizable and worthy of publication.

There are however, substantial problems with the methodology and what was done in reality:

1. mild head injury and (significant) traumatic brain injury were assessed together as one group. These are actually two different pathologies.

2. It seems to be there were few significant traumatic brain injury patients. Actually, in the presentation of the patients, only age and gender are provided. Data concerning the patients' degree of injury can only be extracted from the indicators (which is difficult). It seems from these indicators that around 16 had GCS of 9-13 and 31 had GCS of 3-13…

In this sense, the authors need to describe how many patient records with mild head injury and how many patients with significant traumatic brain injury did they evaluate per site.

Since the actual findings of the survey itself are mainly of local significance, the authors should also concentrate their discussion to issues concerning the methodology (beyond the idea of structured electronic documentation):

1. In what way can the survey be improved (if at all)

2. In what way can compliance be improved.

6. PLOS authors have the option to publish the peer review history of their article (what does this mean?). If published, this will include your full peer review and any attached files.

Reviewer #1: Yes: Itamar Ashkenazi

---

## [Author Response · Author response to Decision Letter 0]

4 Jan 2020

All reviewer comments have been addressed and are outlined in the file "Response to Reviewers" uploaded separately.

---

## [Editor Report · Decision Letter 1]

23 Jan 2020

Guideline Adherence in the Management of Head Injury in Australian Children: A Population-Based Sample Survey

PONE-D-19-26245R1

Dear Dr. Braithwaite,

We are pleased to inform you that your manuscript has been judged scientifically suitable for publication and will be formally accepted for publication once it complies with all outstanding technical requirements.

With kind regards,

Itamar Ashkenazi

Academic Editor

PLOS ONE
---

## [Editor Report · Acceptance letter]

28 Jan 2020

PONE-D-19-26245R1 

Guideline Adherence in the Management of Head Injury in Australian Children: A Population-Based Sample Survey 

Dear Dr. Braithwaite:

I am pleased to inform you that your manuscript has been deemed suitable for publication in PLOS ONE. Congratulations! Your manuscript is now with our production department. 

With kind regards,

on behalf of

Dr. Itamar Ashkenazi 

Academic Editor

PLOS ONE